# Analysis of Microbiota and Mycobiota in Fungal Ball Rhinosinusitis: Specific Interaction between *Aspergillus fumigatus* and *Haemophilus influenza*?

**DOI:** 10.3390/jof7070550

**Published:** 2021-07-10

**Authors:** Sarah Dellière, Eric Dannaoui, Maxime Fieux, Pierre Bonfils, Guillaume Gricourt, Vanessa Demontant, Isabelle Podglajen, Paul-Louis Woerther, Cécile Angebault, Françoise Botterel

**Affiliations:** 1Unité de Parasitologie-Mycologie, Département de Prévention, Diagnostic et Traitement des Infections, APHP, GHU Hôpitaux Universitaires Henri-Mondor, 94010 Créteil, France; sarah.delliere@aphp.fr (S.D.); cecile.angebault@aphp.fr (C.A.); 2Unité de Parasitologie-Mycologie, Hôpital Saint-Louis, Assistance Publique des Hôpitaux de Paris, Université de Paris, 75010 Paris, France; 3UR DYNAMiC 7380, Faculté de Santé, Université Paris-Est Créteil, 94010 Créteil, France; eric.dannaoui@aphp.fr (E.D.); paul-louis.woerther@aphp.fr (P.-L.W.); 4UR DYNAMiC 7380, Ecole Nationale Vétérinaire d’Alfort, USC Anses, 94700 Maison-Alfort, France; 5Unité de Parasitologie-Mycologie, Département de Microbiologie, Hôpital Européen George Pompidou, APHP, Université de Paris, 75015 Paris, France; 6Département d’Otorhinolaryngologie, Hôpital Européen George Pompidou, APHP, Université de Paris, 75015 Paris, France; maxime.fieux@aphp.fr (M.F.); pierre.bonfils@aphp.fr (P.B.); 7Service d’Otorhinolaryngologie, d’Otoneurochirurgie et de Chirurgie Cervico-Faciale, Centre Hospitalier Lyon Sud, Hospices Civils de Lyon, 69310 Pierre Bénite, France; 8Plate-Forme Genomiques, APHP-IMRB, GHU Hôpitaux Universitaires Henri-Mondor, UPEC, 94010 Créteil, France; guillaume.gricourt@aphp.fr (G.G.); vanessa.demontant@aphp.fr (V.D.); 9Unité de Bactériologie, Département de Microbiologie, Hôpital Européen George Pompidou, APHP, Université de Paris, 75015 Paris, France; isabelle.podglajen@aphp.fr; 10Unité de Bactériologie, Département de Prévention, Diagnostic et Traitement des Infections, APHP, GHU Hôpitaux Universitaires Henri-Mondor, 94010 Créteil, France

**Keywords:** *Aspergillus fumigatus*, *Haemophilus influenzae*, mycobiota, microbiota, microbial interactions, fungal–bacterial biofilm, chronic sinusitis

## Abstract

Fungal ball (FB) rhinosinusitis (RS) is the main type of non-invasive fungal RS. Despite positive direct examination (DE) of biopsies, culture remains negative in more than 60% of cases. The aim of the study was to evaluate the performance/efficacy of targeted metagenomics (TM) to analyze microbiota and mycobiota in FB and find microbial associations. Forty-five sinus biopsies from patients who underwent surgery for chronic RS were included. After DE and culture, DNA was extracted, then fungal ITS1–ITS2 and bacterial V3–V4 16S rDNA loci were sequenced (MiSeq^TM^ Illumina). Operational taxonomic units (OTUs) were defined via QIIME and assigned to SILVA (16S) and UNITE (ITS) databases. Statistical analyses were performed using SHAMAN. Thirty-eight patients had FB and seven had non-fungal rhinosinusitis (NFRS). DE and culture of FB were positive for fungi in 97.3 and 31.6% of patients, respectively. TM analysis of the 38 FB yielded more than one fungal genus in 100% of cases, with *Aspergillus* in 89.5% (34/38). *Haemophilus* was over-represented in FB with >1000 reads/sample in 47.3% (18/38) compared to NFRS (*p* < 0.001). TM allowed fungal identification in biopsies with negative culture. *Haemophilus* was associated with FB. Pathogenesis could result from fungi–bacteria interactions in a mixed biofilm-like structure.

## 1. Introduction

Chronic rhinosinusistis (CRS) is a complex and multifactorial condition, associated with a high prevalence rate of 10.9% [1]. Because of the overlap in symptoms between CRS, acute rhinosinusitis and (non-)allergic rhinitis, CRS diagnosis can be difficult based on symptoms alone. The addition of nasal endoscopy or computed tomography (CT) scan showing signs of mucosal inflammation can help provide a reliable CRS diagnosis [2]. CRS can be classified into CRS with nasal polyps (CRSwNP) and CRS without nasal polyps (CRSsNP). In both cases, the relationship between CRS and asthma is indisputable [3]. Indeed, the prevalence of asthma is around 25% in patients with CRS compared to 5% in the general population [4].

Fungi, which are ubiquitous in our environment, belong to the upper airway microbiota and can be associated with several forms of sinus diseases. In these situations, rather than the fungi itself, it is usually the host immune state that determines the clinical presentation [2]. Fungal CRS represent a wide spectrum of diseases ranging from allergic fungal rhinosinusitis (AFRS), mild forms secondary to colonization (saprophytic fungal infestation and fungal ball), to chronic granulomatous and invasive fungal rhinosinusitis (IFRS). AFRS is a subset of CRSwNP characterized by the presence of eosinophilic mucin with non-invasive fungal hyphae within the sinuses and a type I hypersensitivity to fungi [2]. AFRS accounts for about 5–10% of CRS cases [2]. Fungal balls (FB) are non-invasive collections of fungal mycelium obstructing sinuses, while IFRS is almost always associated with an immunocompromised state, of which diabetes (50%) and hematologic malignancy (40%) account for 90% [5]. Other than the immune state, the distinction between invasive and non-invasive fungal CRS depends on the presence or absence of fungal hyphae in the mucosa [1]. FB is a condition that affects mostly adults in their fifth and sixth decade [5,6]. Although FB diagnosis can be incidental, most FB are associated with obstruction symptoms. As shown by Mensi et al., prior endodontic treatment is a risk factor for maxillary FB formation [6]. On imaging, “calcifications” and erosion of the sinus’ inner wall are the two features most predictive for FB [7]. One particularity of sinus FB is that despite fungal hyphae observed on direct examination (DE) of biopsies, mycological culture yields a fungus in less than 40% of cases [8,9,10]. This could be the result of complex microbial interactions with bacterial species that might weaken the fungus and inhibit its growth.

The microbiota could also play a pathogenic role in CRS. However, as the nose and sinuses are not sterile, causality between the microorganisms grown in sinusal cultures and CRS is difficult to establish [2]. Analysis of the bacterial (microbiota) and fungal (mycobiota) diversity over the last decade has yielded significant insight into the role of microbial communities in human diseases [11,12,13]. However, the characterization of the upper respiratory tract mycobiota in CRS has only been performed twice to our knowledge [14,15,16] and been never performed in fungal CRS and FB, specifically. Furthermore, although it has been shown that bacteria and fungi can coexist in CRS’s biofilm [17], the concomitant study of the microbiota and mycobiota by high-throughput sequencing (HTS) in this context has never been carried out. The aim of our study was to describe bacterial and fungal diversity in FB and find possible microbial intra- or trans-kingdom associations.

## 2. Materials and Methods

### 2.1. Patients and Clinical Specimens

A total of 45 samples were collected from 45 patients who underwent functional endoscopic surgery (i.e., middle meatus antrostomy or ethmoidectomy) for CRS from 2015 to 2017 at European Georges Pompidou Hospital, a teaching hospital in Paris, France. During surgery, a biopsy was performed on patients suspected of FB and a sinus aspiration was performed when no FB was present. FB rhinosinusitis (FBRS) was defined by (i) a patient addressed to the hospital for surgical removal of probable FB suspected on clinical and radiological arguments and (ii) a positive direct examination and/or culture of the surgical sample sent to the mycology department (*n* = 38). Non-fungal ball rhinosinusitis (i.e., controls, NFBRS) was defined by (i) patients addressed to the hospital for surgical drainage of a chronic RS with no clinical or radiological argument for a FB (*n* = 7). All patients’ files were reviewed by an ENT surgeon and a mycologist and were classified according to the criteria defined prior to sample processing for the metagenomic study. Patient information regarding age, clinical background, sinusitis localization and use of antifungal/antibiotic were collected and analyzed. This study complies with the ethical and legal requirements of French law (15 April 2019) and the Declaration of Helsinki. The database was officially registered with the French Data Protection Authority (Commission Nationale Informatique et Liberté) (no. 2221215). Written or verbal informed consent from all participants was waived because isolates were collected through routine clinical work and patients’ identifiable information had already been anonymized prior to analysis.

### 2.2. Methods

#### 2.2.1. Direct and Histopathologic Examination and Microbiological Cultures

In the mycology department, direct examination of each sample was performed by Calcofluor-White, May–Grunwald–Giemsa and Gomori–Grocott staining. Culture was performed on 2 Sabouraud media supplemented with gentamicin and chloramphenicol (Biorad, Marnes-la-Coquette, France) incubated at 25 and 37 °C, respectively, for three weeks, one ChromAgar media (Becton Dickinson, Grenoble, France) incubated at 37 °C for 10 days and one Brain–Heart Infusion (BHI) broth incubated at 30 °C for 3 weeks. Concurrently, extra material available was stored at −80 °C before further molecular studies. When concomitantly sent to the bacteriology department (*n* = 21), cultures were performed on Columbia Blood agar with and without colistin and nalidixic acid, Chocolate agar PolyViteX, Drigalsky agar and Schaedler broth (bioMérieux, Lyon, France) under aerobic and anaerobic conditions and in presence of 5% CO_2_. Agar plates and Schaedler broth were incubated at 35 °C for 48 h and 14 days, respectively. Bacterial culture was missing for 24 samples because the samples had only been sent to the mycology department.

#### 2.2.2. DNA Extraction Protocol for High Throughput Sequencing

Specimens were retrieved from the −80 °C samples for DNA isolation. A maximum of 250 mg of the sample was pre-treated prior to automated DNA extraction. Sample was submitted to mechanical lysis with 1.4 mm glass beads during two 60 sec cycles at 6400 rpm on a MagNA Lyser instrument (Roche, Mannheim, Germany) and enzymatic lysis with proteinase K. The remainder of the extraction process was automated on QIAsymphony with the DSP DNA midi kit (Qiagen, Hilden, Germany) with protocol VB400 default IC. Non-template DNA isolation control performed by passing PCR-grade water through the same extraction process was included. DNA extracts were stored at −20 °C until amplification.

#### 2.2.3. Library Preparation and Sequencing

Amplicon libraries targeting the V3–V4 16S hyper-variable regions were prepared for bacteria using primers 341 and 785 without degenerated nucleotides [18]. For the ITS1 or ITS2 regions of fungi, amplicon libraries were prepared using primer pairs ITS1F/ITS2 and ITS3/ITS4, respectively [19]. Amplicon PCR, PCR products, purification and sequencing were prepared as previously described by Angebault et al. [20]. The libraries were sequenced using the MiSeq Reagent Kit V2 500-cycle kit (2 × 250) (Illumina) on an Illumina MiSeq platform (Illumina, Evry, France). Raw data are available on the NCBI website (Bioproject PRJNA 701550). One negative control per extraction batch, consisting of DNA-free water, was submitted to the same extraction protocol. Negative controls were pooled for amplification and sequencing.

#### 2.2.4. Taxonomic Assignment, Diversity

After trimming the barcode and adapter sequences from the reads, paired-ends reads were merged, then filtered, according to Bokulich et al. [21], within QIIME2 [22]. Sequences were denoised with Deblur [23] with a length of 250 pb for 16S and 200 pb for ITS. Amplicon sequence variants (ASVs) (that will be further called operational taxonomic units, OTUs) were assigned to SILVA (v132) [24] for bacterial reads (16S) and UNITE (v8.0, released 18 November 2018) [25] for fungal reads (ITS1 and ITS2) using vsearch [26] with a minimal query coverage set to 70%. Databases Mycobank [27], FungiBank (http://fungibank.pasteur.fr, accessed on 7 July 2021), NCBI RefSeq, Greengenes [28], RDP [29] and EzBioCloud [30] (http://www.ezbiocloud.net, accessed on 7 July 2021) were used to manually assign and verify OTUs. Reads with ≥98.7 and ≥94.5% homology and 0.0 e-value were considered for identification at species and genus level, respectively [31]. Data analysis was further performed by SHAMAN (http://shaman.pasteur.fr, accessed on 7 July 2021) [32]. Rarefaction curves were computed to evaluate the quality of the taxonomic diversity assessment. Diversity indexes (Shannon, Simpson and Inverse Simpson) were calculated to compare the homogeneity of the samples in terms of microbiota and mycobiota composition.

#### 2.2.5. Statistical Analysis

Read count normalization was performed using DESeq2 normalization method [33]. The generalized linear model (GLM) implemented in the DESeq2 R package was applied to detect differences in abundance of taxa between fungal balls and controls [34]. The latter was computed, including the patient, type of sinusitis (FB versus control) and the amplification target as main effects. Resulting *p*-values were adjusted according to the Benjamini and Hochberg procedure [35]. A relative abundance cut-off > 1% of total reads and >5% per sample was used for graphical display of the results. Clinical characteristics of the patients and microbiological data were reported as percentage, mean and standard deviation (SD) if variable followed a Gaussian distribution or median and interquartiles (Q1–Q3) otherwise. Univariate analyses used Fisher’s exact, Chi-2, Mann–Whitney or Student’s test depending on the variable type as appropriate. *p* < 0.05 (two-tailed) was considered statistically significant. Statistical analyses were performed using Prism software v8.0.

## 3. Results

### 3.1. Patient Characteristics

A total of 38 patients underwent surgery for FB removal and seven patients underwent surgical drainage of NFBRS (i.e., controls). Patients with NFBRS had nasal polyposis (*n* = 3), intra-sinus foreign body (suspected tumor or dental paste) (*n* = 3) and chronic rhinosinusitis without any specific etiology (*n* = 1). Mean age of the patients was 59 years (±13) and 58.3% (28/45) were female. Three patients were solid organ transplant recipients (heart, *n* = 2; kidney, *n* = 1) and had risk factors for invasive fungal disease (IFD). No patients received antibiotics or antifungals in the 7 days before surgery and sampling. Two patients with FB received antifungal treatment after surgery. The first patient had a suspicion of IFD secondarily ruled out after histologic examination reanalysis by experts. The second one was considered at risk of IFD (heart transplant) and was treated for probable IFD despite any evidence of tissue invasion during histologic examination. Table 1 shows characteristics of patients with FB compared to patients with NFBRS. No other significant differences were found between the two groups; in other words, the patients are considered comparable in age, sex, type of sinusitis, immune status and results of mycological, bacteriological or histological analyses for the remainder of the statistical analyses.

### 3.2. 16S, ITS1 and ITS2 Targeted Amplicon Sequencing Results

After 16S, ITS1 and ITS2 targeted amplicon sequencing, 3,115,698, 3,428,832 and 3,019,299 raw reads were obtained, respectively. The trimming process led to 1,623,694, 289,417 and 1,213,482 reads, with a median of 40,677, 1055 and 23,632 reads/sample for 16S, ITS1 and ITS2, respectively (Table 2). Finally, 671, 113 and 157 OTUs were obtained for 16S, ITS1 and ITS2. Environmental control 16S amplification yielded mostly *Pseudomonas migulae*, *Corynebacterium* sp., *Anaerobacillus* sp. and *Ochrobactrum* sp. Environmental control ITS1 and ITS2 amplification yielded *Sarocladium kiliense* and *Byssochlamys* sp. The results of environmental controls were considered as background signals and further used to detect samples potentially amplifying and sequencing background DNA only. The comparison of taxonomic distribution between environmental controls and samples was performed using a non-parametric Spearman matrix of correlation. Samples with a Spearman rank correlation coefficient above 0.7 were reanalyzed or not included in the study.

### 3.3. Comparison of ITS1 and ITS2 Regions for Mycobiota Analysis

Table 2 summarizes, for ITS1 and ITS2 targets, the number of reads/sample before and after trimming of raw data and the number of OTUs and taxa at genus level computed from trimmed data. Number of reads/sample prior trimming, number of OTUs/sample and number of total taxa were similar for both targets. On the contrary, the number of reads/sample after trimming varied significantly according to the ITS target with a mean of 5967 (±11,231) reads/sample for ITS1 versus 25,374 (±17,969) reads/sample for ITS2 (*p* < 0.001). Among FB specimens, 97.4% (37/38) produced ITS2 amplicons with at least 5000 fungal sequences. In contrast, only 26.3% (10/38) produced ITS1 amplicons with at least 5000 fungal sequences (*p* < 0.001). Overall, 48 and 57 taxa were identified at the genus level using ITS1 or ITS2, respectively. In the 10 most represented taxa identified with ITS1 and ITS2, eight were in agreement (*Aspergillus*, *Scedosporium*, *Hormographiella*, *Malassezia*, *Schizophyllum*, *Oxysporus*, *Sarocladium* and *Itersonilia*). Normalized relative abundances of these eight taxa are represented in Figure 1. With ITS1 analysis, the 9th and 10th most represented taxa were *Penicillium* and *Symmetrospora* (a new genus gathering species previously assigned to *Rhodotorula* and *Sporobolomyces* genera). These two taxa were found with ITS2 as the 11th and 51st most represented taxa, respectively. On the other hand, *Nakaseomyces* (comprising *Candida glabrata* and related species) was the ninth most represented taxa using ITS2 but was not detected using ITS1 and *Neoascochyta* (environmental and plant pathogenic *fungi*) was the 10th most represented taxa with ITS2 but the 24th with ITS1. 

Mean α-diversity indices, Shannon, Simpson and inverse Simpson were 2.08 (±0.47), 0.41 (±0.13), 0.23 (±0.07) and 1.60 (±0.40) for ITS1 and 1.58 (±0.36), 0.21 (±0.12), 0.11 (±0.06) and 1.29 (±0.24) for ITS2. They showed a narrower diversity with ITS2 despite a higher number of OTUs (Appendix A). At the phylum level, the diversity profiles generated with primers targeting ITS1 showed an increased proportion of *Basidiomycetes* over *Ascomycetes* compare to ITS2 (*p* < 0.001) (Figure 1). Considering these findings, further analysis was performed on the ITS2 dataset.

### 3.4. Analysis of Microbial Diversity in Fungal Ball (FB) and Non-Fungal Ball Chronic Rhinosinusitis (NFBRS)

A total of 133 bacterial and 57 fungal taxa at the genus level were found in the samples. A mean of 15 (range; 3–40) bacterial taxa and 5 (range; 1–30) fungal taxa were found per sample. Predominant bacterial taxa (relative abundance > 1% in more than 25% of samples) were *Haemophilus*, *Pseudomonas* and *Staphylococcus* (Appendix A). Predominant fungal taxa were *Aspergillus* and *Malassezia* (Appendix A). OTUs assigned to the genera *Haemophilus*, *Pseudomonas*, *Staphylococcus* and *Streptococcus* for bacteria and *Aspergillus* for fungi were assigned at species or section level when possible (Appendix A). Among the *Haemophilus* genus, 86.3% of reads were *Haemophilus influenzae* and 13.6% could not be further assigned. Among *Pseudomonas*, 57.7, 23.4 and 15.6% were assigned to *Pseudomonas aeruginosa*, *Pseudomonas mossellii* (mostly represented in one patient) and *Pseudomonas migulae* (mostly represented in the environmental control and found at low rate in samples), respectively. Only 1.7% of *Pseudomonas* reads remained unassigned at the species level. Among *Staphylococcus,* the majority of OTUs (91.9%) were assigned as *Staphylococcus aureus* and 73.1% of OTUs assigned as *Streptococcus* belonged to the *milleri* group. Among *Aspergillus*, 80.8, 15.4 and 3.7% were further assigned to sections *Fumigati*, *Flavi* and *Nidulantes*, respectively.

A significant difference in bacterial (*p* = 0.003) and fungal (*p* = 0.001) microbiota between FB and NFBRS was shown by permutational multivariate ANOVA (PERMANOVA) at the genus level according to the Bray–Curtis dissimilarity metric. Results are presented as a principal coordinate analysis matrix in Figure 2. Bacterial and fungal taxonomic diversity of taxa with a relative abundance > 1% of total reads and >5% per sample are detailed in Figure 3A (read counts) and Figure 3B (relative abundance). Taxonomic diversity of all bacterial and fungal taxa is available in Appendix A.

Focusing on bacterial diversity, *Haemophilus* was the predominant taxon representing 30.3% of total bacterial reads. This taxon was present in 47.3% (18/38) of FB with over 1000 reads per samples versus 0% (0/7) in NFBRS. Regarding fungal diversity, *Aspergillus* was the predominant taxa in 89.5% (34/38) of FB but was present in only 1/7 (14.2%) of NFBRS. The four *Aspergillus*-free FB showed a majority of *Sarocladium* (patient 8), *Hormographiella* (patient 27), *Scedosporium* (patient 31) or a variety of 30 taxa dominated by *Malassezia* (patient 15). In NFBRS, very few fungal reads were detected with a mean read per sample of 6350 (±8678) compared to 30,699 (±15,703) in FB (*p* < 0.001). Reads were mostly assigned to *Malassezia* except for the substantial number of 9998 reads of *Aspergillus* section *flavi* in patient 41 (relative abundance = 81.7%). The *Malassezia* taxon was found in 68.2% (30/38) of FB samples and 57.1% (4/7) of NFBRS with a mean relative abundance per sample of 2.3 (range: 0–38.0%) and 38.0% (range: 0–99.3%), respectively (*p* = 0.23). *Haemophilus* and *Aspergillus* were confirmed to be significantly more abundant in FB than NFBRS by a generalized linear model with *p* < 0.001 (Figure 4). No other bacterial taxa were found to be significantly overrepresented in FB compared to NFBRS. Among FB, there were no significant differences observed with a PERMANOVA test of bacterial or fungal diversity profiles according to patient gender, localization of sinusitis and bacterial and fungal culture.

## 4. Discussion

Fungal and bacterial diversity of sinus samples from 45 patients with FBRS (*n* = 38) and NFBRS (*n* = 7) were analyzed by targeted metagenomics. *Aspergillus* and *Haemophilus* were the main fungal and bacterial genera identified in FBRS. From a more technical perspective, we have confirmed the superiority of using ITS2 over ITS1 as a target for the study of fungal diversity in this context. To our knowledge, this is the first study to carry out HTS of both bacterial and fungal diversity in FB by targeted metagenomics.

### 4.1. Targeting ITS1 or ITS2 to Study Fungal Diversity

Microbial diversity studies are greatly influenced by the method used and their results depend on the ability to extract, amplify, sequence and analyze each microbial genus [36]. The National Institutes of Health (NIH) Human Microbiome Project intends to standardize data resources and new technological approaches to enable such study to be undertaken broadly in the scientific community [37]. However, data focusing on the analysis of fungal communities remain scarce as fungi may be more difficult to manipulate and analyze, particularly due to the difficulty of extracting the fungal cell wall [20]. Controversies remain regarding the selection of markers for sequencing and that is why we intended to compare ITS1 and ITS2 sequencing in our own settings. ITS1 amplification yielded a high proportion of non-specific reads, mostly human, leaving five time less reads per sample than ITS2 amplification. This can be explained by the use of ITS1-F universal primer shown to match human DNA [38], although recommended for the identification using pure fungal DNA [39]. The lower number of reads obtained with ITS1 yielded a higher diversity than ITS2 regardless of the diversity index used. This may be due to (i) a biased analysis after normalization of a reduced number of reads and/or (ii) an actual better representation of fungal diversity with increased representation of *Basidiomycota* taxa. Some studies compared ITS1 and ITS2 for fungal profiles [40,41,42,43,44]. Some authors found ITS2 to be more suitable for revealing richness [40] while others reported that ITS1 was probably the best choice for the study of fungal and eukaryotic species [43]. A recent in silico study revealed that the fungal diversity and richness might be overestimated when using ITS1. Moreover, the clustering and taxonomy detected using ITS2 were more similar to those obtained with the whole ITS region rather than with ITS1 alone [44]. This can be explained by the ITS1 region having evolved more rapidly and having a more variable length than ITS2 [45]. The shorter length, lower GC content variation and greater taxonomic information content of ITS2 probably makes it more suitable than ITS1 for HTS studies of fungi. For this reason and because of a much higher number of fungal reads obtained with this marker we performed our analysis using ITS2. The latest reports, unavailable at the time we designed our study, now recommend targeting the ITS2 subregion by using the degenerated gITS7ngs and ITS4ngs primers, while we used non-degenerated ones [36].

The possibility or necessity to validate metagenomics data using an independent technique such as quantitative PCR (qPCR) is a matter of discussion in progress among the community, similarly to what is performed in transcriptomic analyses [46]. In that case, expression of the gene of interest and control genes are measured against defined standards, usually housekeeping genes’ expression, known to be consistently expressed in all cells and despite varying conditions [46]. In targeted metagenomics, qPCR could be used to quantify the fungal and bacterial biomass of species of interest (i.e., *Aspergillus* and *Haemophilus* in the present study) in order to obtain more comparable results among samples. However, because fungal species differ markedly in number of rRNA operon copies per genome, the performance of qPCR is limited [36]. However, targeting a unique copy gene would significantly decrease the sensitivity of the method. To the best of our knowledge, no such validation was carried out in a previous metagenomic study on fungal diversity.

### 4.2. Fungal Diversity in FBRS

Analysis of the fungal mycobiota found *Aspergillus* to be the predominant genus in 89.5% of FB, which correlates with previous studies, such as Morio et al.’s work, who identified *Aspergillus* in over 77.4% of sinus FB by targeted PCR [8,9,10]. Patients were classified as FB on clinical, radiological and surgical criteria associated with a positive direct examination, showing fungal hyphae and/or a positive culture. In FB, only one patient (patient 15) had a negative direct examination with a positive *A. fumigatus* culture. This 53-year-old patient had pansinusitis secondary to a suspected FB on the CT scan. Amplification of 16S yielded 98.6% of *S. aureus* (Figure 3A). ITS2 amplification yielded only 11,496 reads belonging to 30 fungal taxa with a majority of *Malassezia* reads, a difficult-to-culture commensal yeast (Figure 3B). The variety of fungal taxa may reflect transient, contaminant aerial fungi and the presence of *Malassezia* may reflect nasal mycobiota contamination in a sample with a low fungal biomass [47]. This patient was probably wrongly classified in the FB part. Initial criteria for FB classification should thus have been limited to a positive direct examination. Surprisingly, the *Aspergillus* section *Flavi* was found with a significant number of reads in NFBRS (patient 41). This 48-year-old patient, with a history of a right maxillary cystic lesion surgically removed at the age of 20, presented with chronic rhinosinusitis of dental origin of the same sinus. The detection of *Aspergillus* section *Flavi* in this patient using HTS may be explained by a transient or contaminant fungus in the sample. Indeed, fungal spores are found in the environment and may be inhaled or may contaminate a culture or a sample before extraction. 

The small number of FB included and our choice of NFBRS as a comparison group in this study are the main limits of our study. Indeed, these NFBRS samples were addressed to the mycology lab due to a suspicion of fungal etiology, which constitutes a first bias. In addition, the sampling method for these samples was different from FB. NFBRS samples were pus aspirates and not biopsies. In a prospective study, another control could be proposed, such as sinus biopsies performed for cancer research in patients without prior suspicion of fungal infection.

### 4.3. Hypothesis to Explain the Poor Positive Culture Rate of FBRS Samples

*Aspergillus*, the predominant fungus found in FBRS, is known to grow easily on routine culture media. This raises questions regarding the absence of growth in a majority (60.6–75.0%) of FB samples in our study, as well as in previous works [8,9,10]. A first hypothesis is that interactions between *Aspergillus* and bacterial species could weaken the fungus and inhibit its growth. In their study, based on culture methods, Zhang et al. showed that pathogenic bacteria were associated in more than 75% of FB. They observed that the presence of *Pseudomonas aeruginosa* was significantly associated with altered and fragmented mycelium and that *Aspergillus* growth was inhibited in mixed infection [48]. They therefore hypothesized that in the presence of certain bacteria involved in FB, *Aspergillus* may be weakened, and its growth may be compromised in vitro. However, they did not identify a particular profile or specific bacterial taxa associated with culture-positive FB compared to culture-negative FB [48]. Of note, in non-fungal CRS and particularly in polypoid forms, *S. aureus* has been described as being involved in the pathogenesis of the disease. Many characteristics of *S. aureus*, such as its ability to regulate innate and adaptative immunity, to disrupt tissue barrier function or to promote impaired mucociliary clearance, may favor its role in CRS pathogenesis [49,50]. However, it was not specifically associated with FBRS in our study, unlike *Haemophilus*. Another hypothesis could be that the fungi are embedded in a biofilm-like structure, thus living in a hypometabolic state preventing secondary growth in in vitro conditions [51]. The concept of “viable, but not culturable” (VBNC) organisms has been well described in bacterial biofilms [52] and in yeasts [53,54], but never in filamentous fungi such as *Aspergillus*, to our knowledge. However, in mixed biofilms, histological studies on human tissue or animal models have confirmed the presence of aggregates of hyphae embedded in an extracellular matrix [55,56]. This validates the presence of Aspergillus biofilms, which, in the presence of bacteria, have recently become a topic of interest [57,58,59]. Besides the VBNC hypothesis, biofilms could also contribute to the virulence of *Aspergillus* by mediating adherence to host cells [55] and enhancing its resistance to antifungals and the immune system [60,61].

### 4.4. Could FBRS Result from Aspergillus–Haemophilus Microbial Interaction?

Using metagenomics, *Haemophilus* was found to be significantly more abundant in FB compare to NFBRS. Therefore, we hypothesize that *Aspergillus*–*Haemophilus* interactions could play a definite role in FB pathogenesis. *Haemophilus influenzae*, particularly, is known to be part of the oropharyngeal microbiota and can be responsible for otitis, sinusitis, persistent bronchitis and COPD exacerbations [62]. A longitudinal metagenomic study by Mackenzie et al. did not find a significant amount of *Haemophilus* in healthy bacterial sinusal microbiota. These authors did not find *Haemophilus* to be persistent throughout annual and seasonal changes [63]. Colonization and pathogenesis of *Haemophilus* on host tissue relies on different strategies. Among those, it has already been observed in vitro that *Haemophilus* inhibits the mucociliary clearance system [64]. This strategy could certainly benefit *Aspergillus* in sinuses, allowing inhaled spores to stay in airways and germinate. Except for the study led by Zhang et al. (2015), who performed bacterial cultures on FB, bacterial diversity in FB has not yet been well explored. Zhang et al. mainly found *S. aureus* and *P. aeruginosa* associated with FB, as in our study, but they did not find, as we did, an elevated prevalence of *Haemophilus* [48]. However, *Haemophilus* are bacteria that are sometimes difficult to grow, requiring specific vitamins-enriched media [62]. Working on the sequencing of three ethmoidal sinus samples, the same authors mainly found reads from *Aspergillus* and *Haemophilus* [65]. Interestingly, a study investigating allergic fungal RS (*n* = 4) and chronic RS (*n* = 7) observed polymicrobial biofilms with *Haemophilus* and fungal elements in 82% of sinus biopsies [17]. Although these types of sinusitis are different from FB, these results confirm the possible co-existence of *Haemophilus* and *Aspergillus* in a biofilm structure that could benefit both microorganisms in terms of persistence and growth in the sinuses. Another study from Boase et al. studied mixed biofilm formation in an in vivo model of fungal sinusitis in sheep [66]. They described a toxin inhibiting the ciliary movement of epithelial cells that allowed the formation of fungal biofilm. They also observed that fungal biofilm could form only after co-inoculation with *S. aureus*, *Staphylococcus epidermidis* or *P. aeruginosa*, but surprisingly not with *Haemophilus*. However, it was later shown that the *Haemophilus* ATCC 49247 strain used by Boase et al. was able to induce biofilm formation [67]. Our study does not allow us to conclude whether the co-existence of *Haemophilus* and *Aspergillus* in FB is coincidental, or if a true interaction between the two organisms takes place. It is possible that fungi and bacteria cooperate to enhance defense mechanisms toward the host and to facilitate access to nutrients by specific exchanges [52]. FISH analysis of a fine cross-section of FB with specific *Aspergillus* and *Haemophilus* probes could help confirm their co-existence, embedded in a common extracellular matrix. Then, in vitro studies would be needed to investigate possible interactions between *Aspergillus* and *Haemophilus*.

## 5. Conclusions

FB are specific entities among chronic rhinosinusitis. Targeted metagenomics confirms incrimination of *Aspergillus* spp. despite a negative culture in most cases. More interestingly, our study revealed the concomitant presence of *Haemophilus* in FB, underlining the importance of sending surgical samples to both clinical bacteriology and mycology laboratories regardless of the obvious fungal etiology. Our study highlights the importance of evaluating inter-kingdom interactions to improve the pathophysiological understanding of entities primarily described as solely fungal, such as sinus FB. Although correlation does not indicate causation, such results stand as a reasonable starting point to further investigate inter-kingdom interactions between *Aspergillus fumigatus* and *Haemophilus* in chronic fungal rhinosinusitis of FB type.

## Figures and Tables

**Figure 1 jof-07-00550-f001:**
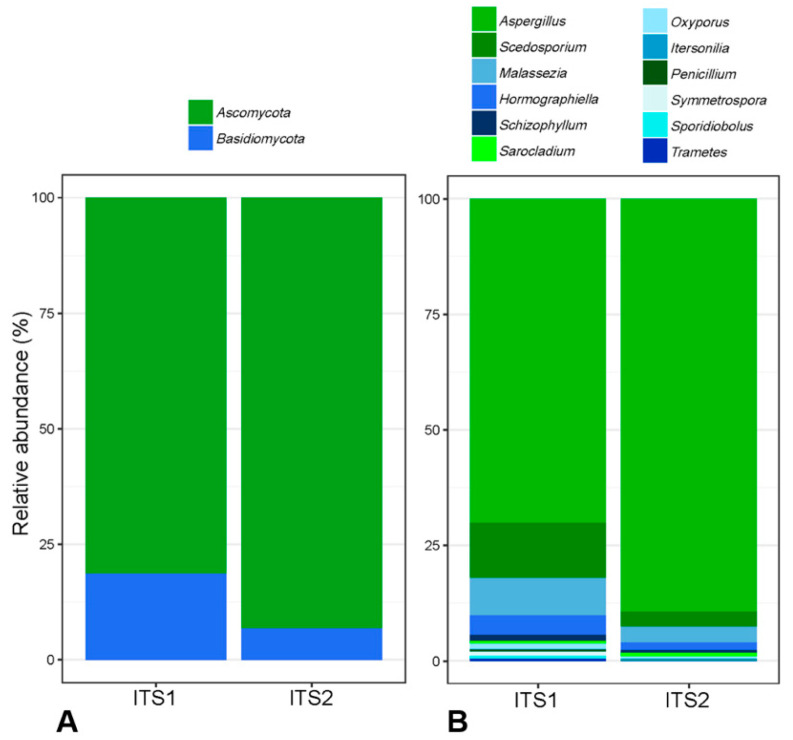
Normalized relative abundances of fungal taxa detected by ITS1 or ITS2 targeted sequencing analysis of all samples at (**A**) phylum level and **(B**) genus level. Distribution of phyla and genera are both statistically different (*p* < 0.0001).

**Figure 2 jof-07-00550-f002:**
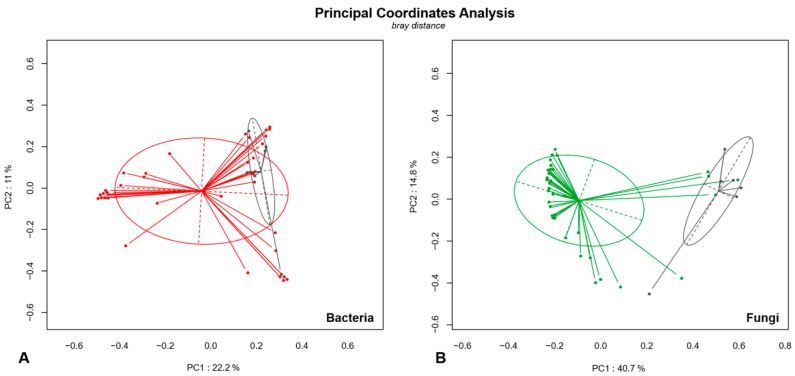
Principal coordinates analysis (PCoA) of bacterial (**A**) and fungal (**B**) diversity according to Bray–Curtis dissimilarity index. Fungus ball (FB) samples are represented in red/green (**A**/**B**) and non-fungal ball rhinosinusitis (NFBRS) are represented in grey. A significant difference in bacterial (*p* = 0.003) and fungal (*p* = 0.001) microbiota between FB and NFBRS was shown by permutational multivariate ANOVA (PERMANOVA) at genus level according to Bray–Curtis dissimilarity metric.

**Figure 3 jof-07-00550-f003:**
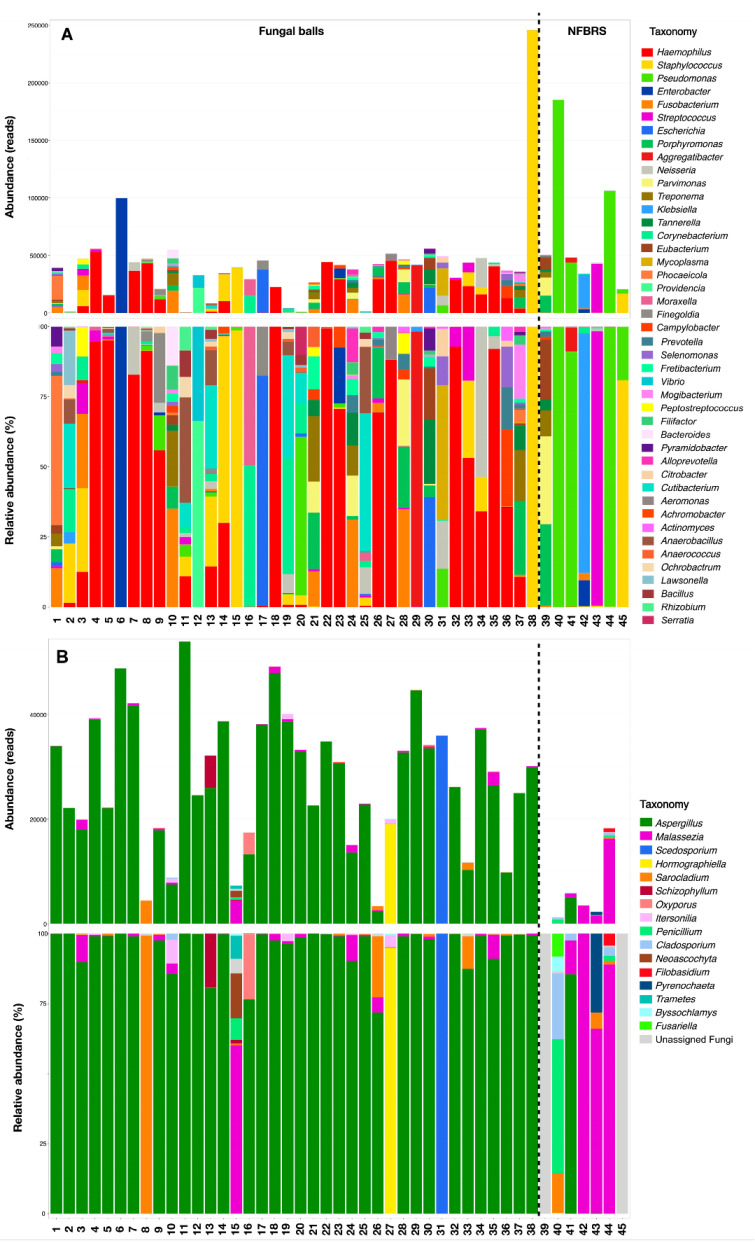
Bacterial (**A**) and fungal (**B**) diversity represented in number of reads and relative abundance for taxa found with a relative abundance > 1% of total reads and >5% of reads per sample.

**Figure 4 jof-07-00550-f004:**
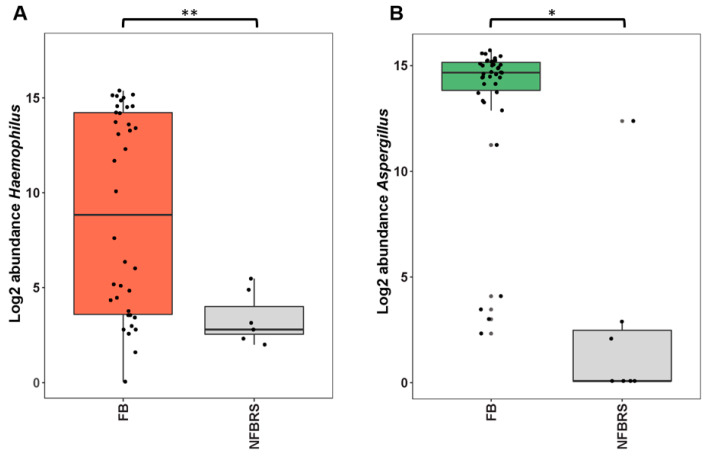
Boxplot of log2 abundances of *Haemophilus* (**A**) or *Aspergillus* (**B**) taxa detected with significantly different abundances between fungus balls (FB) and non-fungal rhinosinusitis (NFBRS). * *p* < 0.05, ** *p* < 0.001.

**Table 1 jof-07-00550-t001:** Characteristics of patients with fungal ball rhinosinusitis (FB) and non-fungal ball rhinosinusitis (NFBRS).

	FB	NFBRS	*p*
Age (+/− SD) (years)	60 (±13)	55 (±14)	0.45
Sex ratio H/F	0.58	0.75	0.99
**Type****of****sinusitis***n* (%)			0.18
Maxillary	24 (63.2)	3 (42.9)	
Frontal	1 (2.6)	1 (14.3)	
Ethmoidal	1 (2.6)	1 (14.3)	
Sphenoidal	7 (18.4)	0 (0)	
Pansinusitis	5 (13.2)	2 (28.6)	
Immunocompromised status	2 (5.1)	1 (14.3)	0.41
**Mycology**			
Positive direct examination	37 (97.3)	0 (0)	<0.001
Positive culture	12 (31.6)	0 (0)	0.16
**Bacteriology**			
Culture performed	14 (36.8)	7 (100)	<0.001
**Histology**			
Performed	26 (68.4)	5 (71.4)	-
Hyphae observed	20 (52.6)	0 (0)	-

FB: fungal ball; NFBRS: non-fungal ball rhinosinusitis; NS: not significant; -: not performed.

**Table 2 jof-07-00550-t002:** Reads, OTUs and taxa found with ITS1 or ITS2 targets (metagenomics targeted analysis).

		ITS1	ITS2	*p*
Before trimming	Minimum reads/sample	47,418	26,384	
	Maximum reads/sample	76,196	67,095	
	Mean reads/sample	129,894	131,564	NS
After trimming	Minimum reads/sample	14	2	
	Maximum reads/sample	35,815	60,335	
	Mean reads/sample	5967	25,374	<0.001
	>5000 reads/FB sample	26.3% (10/38)	97.4% (37/38)	
	OTUs produced	113	157	NS
	Mean OTUs/sample	6	9	
	Taxa (genus level)	48	57	---

OTU: operative taxonomic unit; FB: fungal balls.

## Data Availability

Not applicable.

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
