# Peer review of "Analysis of Microbiota and Mycobiota in Fungal Ball Rhinosinusitis: Specific Interaction between Aspergillus fumigatus and Haemophilus influenza?"

_jof, 2021, doi:10.3390/jof7070550_

Round 1
Reviewer 1 Report
INTRODUCTION
- the authors should clarify how invasive and non-invasive fungal rhinosinusitis exactly are defined.
- Once done that, the authors should clearly define also “fungal ball” RS as a subtype of invasive fungal RS. They basically did that, but I will suggest removing the very approximative clinical description (“…causes nasal obstruction, chronic RS, facial pain and cacosmia.”)
- the second paragraph “One particularity….microbial interactions” should be summarized and such a specific pathogenetic discussion could be included in the discussion. I mean, the authors could just write a sentence to explain the background issue leading to the main study question, namely if “FB could thus be the results of complex microbial interactions.” The supportive or conflicting data about the previous hypotheses should be used to develop the discussion.
- Conversely, in the introduction, before the second paragraph, the authors should provide some general, but clear, background on the clinical concepts on chronic RS and its comorbidities and, in detail, fungal rhinosinusitis, including some epidemiological aspects in adults and children. ((Int J Immunopathol Pharmacol. 2010 Jan-Mar;23(1 Suppl):29-31); & Neuroimaging Clin N Am. 2015 Nov;25(4):569-76. doi: 10.1016/j.nic.2015.07.004).
PATIENTS ANS METHODS
- line 82: can you specify the type of surgical approach?
- can you specify the date and number of the ethical approval granted by the hospital IRB?
- I would suggest creating a different subsection for the description of the statistical methods
RESULTS
- Table 1. positive direct examination: FB = 97.3%: 1 patient was negative at the direct examination. Can you comment on this patient? What about his/her culture?
- Table 1. Histology: performed (/not performed). I am not sure if the statistical analysis of this parameter makes sense. Anyway, it is not clear why the histological analysis was not performed in all patients.
- section 3.4.: there are a few of mistyped bacteria. (e.g. PsSoueudomonas)
- I understand that, according to the present analysis, H. influenzae was the most abundant species (“Haemophilus was the predominant taxon representing 30.3% of total bacterial reads). However, it is well known the role of other bacterial species in chronic RS (and, in detail, S. Aureus) and I think it could be interesting making the same analysis for other abundant bacterial species. Is it possible to use the same approach for S. aureus or, at least, to include this aspect in the discussion later? Indeed, S. aureus has a substantial role in chronic RS and, thus, may play some role even in fungal forms (Curr Allergy Asthma Rep . 2019 Mar 11;19(4):21. doi: 10.1007/s11882-019-0853-7.; & Toxins (Basel). 2020 Oct 28;12(11):678. doi: 10.3390/toxins12110678).
DISCUSSION
- As mentioned above, some aspects of the introduction (second paragraph) should be moved in the introduction and some discussion on S. aureus should be included.
- at the beginning of the discussion, the authors should summarize more clearly and with some more details/specifications their main finding, that they are going to discuss in this section.
- The authors should highlight some clear point/issues that they want to discuss. Indeed, this section looks a little dispersive and it is not always easy to understand the point. If needed and appropriate, I think the authors could organize the discussion in different subsections. Actually, I think that would improve a lot the readability of the discussion.
- Aspergillus paragraph: are there any potential role of the host immune response (e.g. IgE-mediate mechanisms, basophil response, etc.) in development the fungal RS and, in detail, in fungal ball RS? If possible, might you shortly discuss this point as well.
- After reading the last 2 sections, I further recommend reorganizing the discussion, including the creation of specific subsections with respective subheadings.
- At the end of the discussion, the authors should discuss the study limitations.
CONCLUSION
- please, provide only clear and concise take home message(s).
REFERENCES
- to be updated and completed according to the comments and recommendations above
Author Response
Reviewer 1
INTRODUCTION
- the authors should clarify how invasive and non-invasive fungal rhinosinusitis exactly are defined.
ACTION: This has been better clarified line 65-69: “Fungal balls (FB) are non-invasive collections of fungal mycelium obstructing sinuses while IFRS is almost always associated with immunocompromised state, of which diabetes (50%) and haematologic malignancy (40%) account for 90% [4]. Other than the immune state, the distinction between invasive and non-invasive fungal CRS depends on the presence or absence of fungal hyphae in the mucosa [1].
- Once done that, the authors should clearly define also “fungal ball” RS as a subtype of invasive fungal RS. They basically did that, but I will suggest removing the very approximative clinical description (“…causes nasal obstruction, chronic RS, facial pain and cacosmia.”)
REPLY: We suppose the reviewer meant “define also fungal ball RS as a subtype of NON-invasive fungal RS” and agree to propose a better definition (see above).
ACTION: We have removed the sentence “causes nasal obstruction, chronic RS, facial pain and cacosmia”
- the second paragraph “One particularity….microbial interactions” should be summarized and such a specific pathogenetic discussion could be included in the discussion. I mean, the authors could just write a sentence to explain the background issue leading to the main study question, namely if “FB could thus be the results of complex microbial interactions.” The supportive or conflicting data about the previous hypotheses should be used to develop the discussion.
ACTION: This paragraph have been reduced to line 74-77: “One particularity of sinus FB is that despite fungal hyphae observed on direct examination (DE) of biopsies, mycological culture yields a fungus in less than 40% of cases [4-6]. This could be the result of complex microbial interactions with bacterial species that might weaken the fungus and inhibit its growth.”
We have transferred the rest of the paragraph to the discussion section (line 393-402). A first hypothesis is that interactions between Aspergillus and bacterial species could weaken the fungus and inhibit its growth. In their study, based on culture methods, Zhang et al. showed that pathogenic bacteria were associated in more than 75% of FB. They observed that the presence of Pseudomonas aeruginosa was significantly associated with altered and fragmented mycelium and that Aspergillus growth was inhibited in mixed infection [7]. They therefore hypothesized that in presence of certain bacteria involved in FB, Aspergillus may be weakened, and its growth may be compromised in vitro. However, they did not identify a particular profile or specific bacterial taxa associated with culture-positive FB compared to culture-negative FB [7].
- Conversely, in the introduction, before the second paragraph, the authors should provide some general, but clear, background on the clinical concepts on chronic RS and its comorbidities and, in detail, fungal rhinosinusitis, including some epidemiological aspects in adults and children. (Int J Immunopathol Pharmacol. 2010 Jan-Mar;23(1 Suppl):29-31); & Neuroimaging Clin N Am. 2015 Nov;25(4):569-76. doi: 10.1016/j.nic.2015.07.004).
REPLY: We thank the reviewer for suggesting additional literature and we read the article by Raz et al. with great interest and cited it. However, we were not able to find the other article that we assume to be Marseglia GL, Caimmi S, Marseglia A, Poddighe D, Leone M, Caimmi D, Ciprandi G, Castellazzi AM. Rhinosinusitis and asthma. Int J Immunopathol Pharmacol. 2010 Jan-Mar;23(1 Suppl):29-31. PMID: 20152076. However, we found other articles from the same team focusing on epidemiological aspects of rhinosinusitis (Castagnoli et al, Expert Rev Respir Med, 2020; Poddighe et al, Respir Med, 2018; Licari et al, Int J Immunopathol Pharmacol, 2014) and we read those with great interest.
ACTION: The introduction have been completely rearrange to meet the reviewer requests and new paragraphs (from line 48 to 70) have been added to better provide general background on the clinical concepts on chronic RS and its comorbidities as well as fungal rhinosinusitis including some epidemiological aspect. The references suggested by Reviewer 1 have been added to the introduction (ref n°3 and 7).
PATIENTS AND METHODS
- line 82: can you specify the type of surgical approach?
ACTION: In order to describe more precisely the type of surgical approach used in our patients we added some clarifications in line 93-95: “A total of 45 samples were collected from 45 patients who underwent functional endoscopic surgery (i.e. middle meatus antrostomy or ethmoidectomy) for chronic rhinosinusitis from 2015 to 2017”
- can you specify the date and number of the ethical approval granted by the hospital IRB?
ACTION: In compliance with French law and regulations, the need for informed consent was waived. The study was nevertheless approved by the Hospital’s Ethics on Committee, and the database was officially registered with the French Data Protection Authority (Commission Nationale Informatique et Liberté) (No.2221215 dated 17/02/2021).
- I would suggest creating a different subsection for the description of the statistical methods
ACTION: We agree with Reviewer 1 and added a statistical subsection in the Material and Methods section (line 167-180).
RESULTS
- Table 1. positive direct examination: FB = 97.3%: 1 patient was negative at the direct examination. Can you comment on this patient? What about his/her culture?
REPLY: Information/explanation regarding this patient is available line 366-375: “In FB, only one patient (patient 15) had a negative direct examination with a positive A. fumigatus culture. This 53-year-old patient had pansinusitis secondary to a suspected FB on the CT-scan. Amplification of 16S yielded 98.6% of S. aureus (Figure 3A). ITS2 amplification yielded only 11,496 reads belonging to 30 fungal taxa with a majority of Malassezia reads, a difficult-to-culture commensal yeast (Figure 3B). The variety of fungal taxa may reflect transient, contaminant aerial fungi and the presence of Malassezia may reflect nasal mycobiota contamination in a sample with a low fungal biomass [41]. This patient was probably wrongly classified in the FB part. Initial criteria for FB classification should thus have been limited to a positive direct examination.”
- Table 1. Histology: performed (/not performed). I am not sure if the statistical analysis of this parameter makes sense. Anyway, it is not clear why the histological analysis was not performed in all patients.
REPLY: Histology in this context was mostly performed in a non-systematic manner by the surgeons depending on personal practice. Some only send the material for fungal analysis ± bacterial analysis since they suspected an infectious disease confirmed by the CT scan. Some surgeons further sent it in order to rule out invasive fungal sinusitis with tissue invasion.
ACTION: We agree with Reviewer 1 and have deleted the statistical analysis regarding histology since it is indeed not highly informative in this particular case.
- section 3.4.: there are a few of mistyped bacteria. (e.g. Pseudomonas)
ACTION: This have been corrected.
- I understand that, according to the present analysis, H. influenzae was the most abundant species (“Haemophilus was the predominant taxon representing 30.3% of total bacterial reads). However, it is well known the role of other bacterial species in chronic RS (and, in detail, S. Aureus) and I think it could be interesting making the same analysis for other abundant bacterial species. Is it possible to use the same approach for S. aureus or, at least, to include this aspect in the discussion later? Indeed, S. aureus has a substantial role in chronic RS and, thus, may play some role even in fungal forms (Curr Allergy Asthma Rep . 2019 Mar 11;19(4):21. doi: 10.1007/s11882-019-0853-7.; & Toxins (Basel). 2020 Oct 28;12(11):678. doi: 10.3390/toxins12110678).
REPLY: This is indeed interesting and we thank the Reviewer 1 for his suggestion of articles regarding the role of S. aureus in chronic RS. Regarding our bio-informatic analysis, we performed the same analysis for all bacterial genera but only Haemophilus was found to be significantly associated to FB compare to NFBRS. We did not find that S. aureus was associated to FB compared to NFBRS in our study. Regarding the role of S. aureus, we discussed it in the discussion (line 441-444). “Another study from Boase et al studied mixed biofilm formation in an in vivo model of fungal sinusitis in sheep [61]. They described a toxin inhibiting the ciliary movement of epithelial cells that allowed the formation of fungal biofilm. They also observed that fungal biofilm could form only after co-inoculation with S. aureus, Staphylococcus epidermidis or P. aeruginosa but surprisingly not with Haemophilus”. However, the literature suggested by the reviewer is definitely relevant and will be further included and discussed.
ACTION: To clarify our result regarding bacteria associated with FB compared to NFBRS, we have added the following sentence line 303-304: “No other bacteria was found to be significantly overrepresented in FB compared to NFBRS.” We have also further discussed the role of Staphylococcus aureus line 402-406: “Of note, in non-fungal CRS and particularly in polypoid forms, S. aureus has been described as being involved in the pathogenesis of the disease. Many characteristics of S. aureus, such as its ability to regulate innate and adaptative immunity, to disrupt tissue barrier function or, to promote impaired mucociliary clearance may favor its role in CRS pathogenesis [42,43]. However it was not specifically associated to FB in our study, unlike Haemophilus.”
DISCUSSION
- As mentioned above, some aspects of the introduction (second paragraph) should be moved in the introduction and some discussion on S. aureus should be included.
REPLY: This have been done as recommended by the reviewer. See answers above.
- at the beginning of the discussion, the authors should summarize more clearly and with some more details/specifications their main finding, that they are going to discuss in this section.
ACTION: The first part of the discussion has been more clearly summarized as follow: “Fungal and bacterial diversity of sinus samples from 45 patients with FBRS (n=38) and NFBRS (n=7) were analysed by targeted metagenomics. Aspergillus and Haemophilus were the main fungal and bacterial genera respectively identified in FBRS. Furthermore, in a more technical perspective we have confirmed the superiority of using ITS2 over ITS1 as target for the study of fungal diversity in this context. To our knowledge, this is the first study to carry out HTS of both bacterial and fungal diversity in FB by targeted metagenomics.”
- The authors should highlight some clear point/issues that they want to discuss. Indeed, this section looks a little dispersive and it is not always easy to understand the point. If needed and appropriate, I think the authors could organize the discussion in different subsections. Actually, I think that would improve a lot the readability of the discussion.
REPLY: We agree with the reviewer that the discussion develops many points and that subheadings would make it easier for the reader to understand and read.
ACTION: Thus, we have reorganized our discussion by adding the following subheadings:
- line 318, 4a: “Targeting ITS1 or ITS2 to study fungal diversity
- line 3614b: “Fungal diversity in FBRS”
- line 390 4c:. “Hypothesis to explain the poor positive culture rate of FBRS samples”
- line 418 4d: “Could FBRS result from Aspergillus-Haemophilus microbial interactions?”
- Aspergillus paragraph: are there any potential role of the host immune response (e.g. IgE-mediate mechanisms, basophil response, etc.) in development the fungal RS and, in detail, in fungal ball RS? If possible, might you shortly discuss this point as well.
REPLY: To our knowledge, fungal ball risks factors never included asthma or allergic conditions. We believe trying to bridge IgE-mediated mechanisms and fungal ball may mislead the readers as this is related to a complete different entity that are allergic fungal rhinosinusitis and that it may be off topic in our case.
ACTION: We have detailed in the introduction the different types of fungal rhinosinusitis including AFRS for more clarity (line 63-64): “Allergic fungal rhinosinusitis (AFRS) is a subset of polypoid chronic rhinosinusitis (CRSwNP ) characterized by the presence of eosinophilic mucin with non-invasive fungal hyphae within the sinuses and a type I hypersensitivity to fungi [2].
- After reading the last 2 sections, I further recommend reorganizing the discussion, including the creation of specific subsections with respective subheadings.
ACTION: We agree with the reviewer and added subheadings to clarifiy the discussion section (see answer to reviewer 2).
- At the end of the discussion, the authors should discuss the study limitations.
REPLY: We choose to discuss each of the study limitations in the related subsections, because we thought it might be clearer this way for the reader. For instance, we discussed the issue of the ITS2 primers used in our study at the end of the “ITS1 vs ITS2 target to study fungal diversity” subsection and introduced the new primers recommended by Nilsson et al (Nat Biotechnol 2019). Cf. line 345 “Latest reports, unavailable at the time we designed our study, now recommend targeting the ITS2 subregion by using the degenerated gITS7ngs and ITS4ngs primers while we used non degenerated ones [26]". We also discussed at the end of the paragraph on “Fungal diversity in FBRS” the fact that the main limit of our study was the small number of FB included. Cf. line 382-383 “The small number of FB included and our choice of NFBRS as a comparison group in this study are the main limit of our study.”
CONCLUSION
- please, provide only clear and concise take home message(s).
REPLY: Giving clear take home message per se is difficult from our work since this is a descriptive study using techniques that are still being optimized and standardized by the scientific community. Also, we believe it is important to remain cautious and balanced in the conclusion regarding our findings in order not to fool the community with misleading certainties. In the conclusion section, we aimed at summarizing our main findings and perspectives as clearly as possible. Cf. line 455-459.
REFERENCES
- to be updated and completed according to the comments and recommendations above
ACTION: We updated our list of references and included the articles recommended by the reviewers (ref 3, 7).
Reviewer 2 Report
This is an interesting report on fungal and bacterial taxa being detected in fungal ball rhinosinusitis and these merit publication. However there are some issues that need addressing:
- Use either the term mycobiota or mycobiome or fungal microbiota consistently throughout the manuscript
- Have a consistent color scheme for graphics (Aspergillus should have same color in all figures). In Figure 1 also it is virtually impossible to discern the different genera due to very similar colors being used
- Did the authors use OTUs as written or are these ASVs (exact sequencing variants) as you would expect when using Dblur. If OTUs please, clarify what similarity threshold was used
- BLAST (l.145) is not a database but an algorithm to assign taxonomy to reads. Please clarify which database was used in BLAST.
- Table 1: please give exact p-values instead of blanket statements of “non-significant”.
- Table 1: What does “culture performed” under bacteriology mean: only whether bacterial culture was performed or whether bacteria actually could be cultured? Please also report the results of bacterial culture and describe whether these are similar to the 16S results.
- Table 2 (and all mentions of read numbers in the text, such as 275-285) is mostly irrelevant: the absolute number of reads is not a meaningful metric per se. It is not directly related to the amount of pathogens in a specific sample. I would remove this as there is no meaningful information to be gained. If the authors want to assess fungal/bacterial burden, I would suggest a quantitative method such as 18S/16S qPCR, even if (as the authors correctly discuss) there are cavets to this method too.
- The authors describe differences between ITS1 and ITS2 sequencing, such as the non-detection of C. glabrata in ITS1. I fear this is due to the use of PE250 settings. The expected ITS1 amplicon is typically larger than 500bp for C. glabrata. Without accounting for these lengths , any ITS1 results will give an incorrect taxonomic representation.
- Figure 2: Please add descriptions (“Bacteria”/”Fungi”) as well as color legends to the figure, so that the reader does not have to look for these details in the text below the figure.
- Figure 3: the absolute number of reads does not provide meaningful information. I suggest removing it. Further I suggest using a similar color scheme in all fungal figures
Author Response
This is an interesting report on fungal and bacterial taxa being detected in fungal ball rhinosinusitis and these merit publication. However there are some issues that need addressing:
- Use either the term mycobiota or mycobiome or fungal microbiota consistently throughout the manuscript
REPLY: We agree with the reviewer regarding the fact of using always the same term for the same idea. Thus, we have decided to use mycobiota and remove mycobiome or fungal microbiota.
ACTION: We have modified this across the text.
- Have a consistent color scheme for graphics (Aspergillus should have same color in all figures). In Figure 1 also it is virtually impossible to discern the different genera due to very similar colors being used
REPLY and ACTION: As recommended by the reviewer, we choose a color scheme for graphics and made some changes in our figures (Figure 1, Figure 2 and Figure 4) to improve their readibility.
The pattern chosen was mainly used to be consistent with the color of the two main pathogens found in our study. In Figure 1, Figure 3 and Figure 4, Aspergillus is represented in green and Haemophilus in red.
In Figure 2, mycobiota profiles of FB are (by extension) represented in green, while mycobiota of NFBRS are grey. For bacteria (2A), microbiota profiles of FB are (by extension) in red, while NFBRS microbiota are grey.
In Figure 1, we added another color scheme. Ascomycota taxa are all in various shades of green, while Basidiomycota are depicted in various shades of blue.
- Did the authors use OTUs as written or are these ASVs (exact sequencing variants) as you would expect when using Dblur. If OTUs please, clarify what similarity threshold was used
REPLY : We thank the reviewer for pointing out this error regarding the use of the term OTUs. Using DeBlur, we indeed obtained ASVs. No similarity threshold was thus used for OTUs computing.
ACTION: We corrected that mistake in the methods section (line 153 -154) and and we have replaced in the text and figures ASVs by OTUs for better understanding for the readers
- BLAST (l.145) is not a database but an algorithm to assign taxonomy to reads. Please clarify which database was used in BLAST.
REPLY : We thank the reviewer for pointing out this error, we used the RefSeq database.
ACTION: This has been modified in the method section (line 157).
- Table 1: please give exact p-values instead of blanket statements of “non-significant”.
ACTION: All P-values are now in Table 1.
- Table 1: What does “culture performed” under bacteriology mean: only whether bacterial culture was performed or whether bacteria actually could be cultured? Please also report the results of bacterial culture and describe whether these are similar to the 16S results.
REPLY: Since this is a retrospective study, not all samples were send to the bacteriology department. Some of them were sent to the mycology department only when the diagnosis was obvious for the surgeon.
Because bacterial culture was only performed in less than 50% of cases we felt like describing precisely these results (culture results and comparison with HTS) would complicate our manuscript without adding truly relevant information. However, to answer to the reviewer’s interest, here are some data regarding the results of bacterial cultures vs. HTS. Species identified by culture and HTS were compared when both were available (n=21 for bacterial culture, 46.6% of cases). Bacterial results were considered in agreement with HTS if species identified in culture were found as most abundant taxa in HTS. Partial agreement was defined when species identified in culture corresponded to minor taxa in HTS and discordance/discrepancy if species identified by in culture were not found by HTS.
Results of bacterial culture and HTS were in agreement or partial agreement for 71.4% (15/21) and 19.0% (4/21) of samples respectively and discordant in 9.5% (2/21) of samples. The two discrepancies corresponded to a profile made up of anaerobic bacteria in HTS while culture yielded Klebsiella oxytoca (patient 24) and a profile mainly made up of Pseudomonas and Aggregatibacter in HTS while culture yielded S. aureus (patient 41).
ACTION: We have clarified in the methods section line 126-127 “Bacterial culture was missing for 24 samples because samples were only sent to the mycology department”.
- Table 2 (and all mentions of read numbers in the text, such as 275-285) is mostly irrelevant: the absolute number of reads is not a meaningful metric per se. It is not directly related to the amount of pathogens in a specific sample. I would remove this as there is no meaningful information to be gained. If the authors want to assess fungal/bacterial burden, I would suggest a quantitative method such as 18S/16S qPCR, even if (as the authors correctly discuss) there are cavets to this method too.
REPLY: We agree with the Reviewer that the number of read is not indeed relevant for taxa identification or fungal/bacterial burden analysis. But this analysis is essential to highlight the superiority of ITS1 over ITS2 has the total reads produced and assigned with the same sample extract is significantly different showing the lack of specificity of ITS1 target/primers amplifying human DNA.
- The authors describe differences between ITS1 and ITS2 sequencing, such as the non-detection of C. glabrata in ITS1. I fear this is due to the use of PE250 settings. The expected ITS1 amplicon is typically larger than 500bp for C. glabrata. Without accounting for these lengths , any ITS1 results will give an incorrect taxonomic representation.
REPLY: This is very interesting information and indeed, the trimming of reads <250b and the truncation of other reads at length 250b could be an issue for assigning some fungal taxa with longer (or shorter) ITS regions. We were aware of this issue and therefore tried different thresholds for the for the trimming and truncation process of the reads (300n; 350n). We did not find any difference regarding the assignment quality with ITS1 by changing the threshold. But by using higher thresholds, we encounter another problem since we lost too many sequences with less than 250n, 300n or 350n. Finally, we stuck to with initial threshold of 250n, which seems to be the best balance between having long-enough reads for proper assignation and losing too many shorter reads. However, the Reviewer is right, the length trimming and truncation process can lead to some issues in assigning fungi and we could have discussed this in our discussion but it is already very long and detailed.
- Figure 2: Please add descriptions (“Bacteria”/”Fungi”) as well as color legends to the figure, so that the reader does not have to look for these details in the text below the figure.
REPLY: Indeed the Figure will be more easily readable with this addition
ACTION: Done
- Figure 3: the absolute number of reads does not provide meaningful information. I suggest removing it. Further I suggest using a similar color scheme in all fungal figures
REPLY and ACTION: We agree with the reviewer that this figure is not necessarily informative. However, it allows us to see that some samples, especially for NFBRS, have fungal reads that are sometimes low in absolute numbers; this is why we wanted to keep this figure. On the other hand, we changed or Figures 1, 2 and 4 in order to better respect a color scheme for graphics. The pattern chose was green for Aspergillus (and by extension the mycobiota of FB patients) and red for Haemophilus (and by extension the microbiota of FB patients).
Round 2
Reviewer 1 Report
I have no additional major comments.